# Lactate Thresholds and Performance in Young Cross-Country Skiers Before and After the Competitive Season: Insights from Laboratory Roller-Ski Tests in Normoxic and Hypoxic Conditions

**DOI:** 10.3390/sports13100344

**Published:** 2025-10-03

**Authors:** Jesús Torres-Pérez, Eneko Fernández-Peña, Alexa Callovini, Aitor Pinedo-Jauregi

**Affiliations:** 1Department of Physical Education and Sport, Faculty of Education and Sport, University of the Basque Country (UPV/EHU), 01007 Vitoria-Gasteiz, Spain; 2CeRiSM, Sport Mountain and Health Research Centre, University of Verona, 37129 Rovereto, Italy; 3Sports and Physical Exercise Research Group (GIKAFIT), Department of Physical Education and Sport, Faculty of Education and Sport, University of the Basque Country (UPV/EHU), 01007 Vitoria-Gasteiz, Spain; 4AKTIBOki: Research Group in Physical Activity, Physical Exercise and Sport, Department of Physical Education and Sport, Faculty of Education and Sport, University of the Basque Country (UPV/EHU), 01007 Vitoria-Gasteiz, Spain; 5Bioaraba, Physical Activity, Exercise and Health Group, 01007 Vitoria-Gasteiz, Spain

**Keywords:** altitude, anaerobic threshold, cross-country skiing, hypoxia, lactate

## Abstract

Cross-country (XC) skiing imposes high physiological demands under hypoxic conditions at altitude. Lactate thresholds such as Onset Blood Lactate Accumulation at 4 mmol/L (OBLA4) and lactate plus 1 mmol/L above baseline (Bsln+1.0) are crucial for tracking performance. This study investigates physiological responses in junior XC skiers under normoxic and hypoxic conditions before (PreCs) and after (PosCs) the competitive season. Nine national-level XC skiers performed a Graded Exercise Test (GXT) on a treadmill using roller skis under both normoxic and hypoxic conditions in PreCS and PosCS. Heart rate, slope (treadmill inclination), and lactate thresholds (Bsln+1.0 and OBLA4) were measured. Significant differences were found between PreCs and PosCs under hypoxia for maximum heart rate (*p* < 0.05). Estimated slopes at Bsln+1.0 and OBLA4 were lower under hypoxia compared to normoxia in PreCs (*p* = 0.005, d = −1.29 for Bsln+1.0 and *p* = 0.013, d = −1.06 for OBLA4). In PosCs, a lower impairment effect of hypoxia exposure under slope at OBLA4 was found (*p* = 0.02, d = −0.95). Positive correlations were found between heart rate and slope for Bsln+1.0 and OBLA4 in PreCs under normoxia and hypoxia, becoming stronger at PosCs, especially under hypoxia. Delta values showed that the higher the slope at Bsln+1.0 and OBLA 4 under normoxia was, the greater the decrease between normoxia and hypoxia was. Physiological changes in junior XC skiers after training and competition in normoxic and hypoxic conditions highlight the importance of hypoxic environments for assessing and monitoring performance throughout the season.

## 1. Introduction

Cross-country (XC) skiing is recognized as one of the principal endurance sports globally, with energy production and movement efficiency being critical determinants of performance [1,2]. In XC skiing, athletes must sustain energy production throughout competitions to execute technical movements at high intensity levels. The demands of XC skiing competitions center on skiers’ ability to generate energy, enabling them to traverse snow-covered terrain with skis, tackle uphill sections, and recover on downhill areas, while completing laps over the race distance. XC skiing competitions range from 1.3–1.8 km sprint races to 10–50 km time trials and mass start races, lasting between approximately 3 min and 2 h [3], often conducted at altitudes between sea level and 1800 m above sea level (m.a.s.l.) [4].

To achieve optimal performance, athletes require high values of maximum oxygen consumption (VO_2_max) [5], peak oxygen consumption (VO_2_peak) [4,6,7], anaerobic capacity [4,7], and Gross Efficiency (GE) [1], and the ability to sustain efforts at high VO_2_ values relative to VO_2_peak [2,7,8]. The relative importance of these factors varies with race type; for instance, higher VO_2_ values at Onset Blood Lactate Accumulation of 4 mmol/L (OBLA4) correlate with performance in elite male skiers over race distances from 5 to 30 km (r = −0.829 to −0.964) and in elite female skiers over race distances from 2.5 to 15 km (r = −0.715 to −0.810) [7]. The relative work intensity during a graded exercise test (GXT) at OBLA4 shows significant correlations with the International Federation competition ranking (CINR) [5] and the 15 km skiing time at the National Swedish Championships (15kmSwedNat) [5]. In addition, relative work reached at the threshold of 1 mmol/L above the lowest lactate value (Bsln+1.0) also correlates with CINR (r = −0.699), 15kmSwedNat (r = −0.718) and 30 km skiing times for the Swedish National championships (r = −0.682) [5].

However, specific physiological tests present challenges in linking laboratory test variables to real-world performance in junior race distances. Junior skiers (aged 14 to 15) compete in shorter sprints of 1.0–2.5 km and longer races of 5.0–7.5 km which differ from elite formats. Aerobic capacity, measured as VO_2_peak, is associated with muscle mass increases and maturation status, key determinants of performance in youth XC skiing [9]. VO_2_peak significantly contributes to performance in 3 min time trials for junior male and female XC skiers [10].

Despite well-established physiological factors influencing performance in elite XC skiers, there is a lack of comprehensive evidence explaining performance in junior skiers. Specifically, the role of external and internal variables, measured before and after the competitive season, remains insufficiently explored, leaving gaps in our understanding of how these variables contribute to performance changes across a competitive season in junior XC skiers.

Additionally, considering the impact of altitude exposure under mentioned physiological success variables such as VO_2_max [11], lactate thresholds [12,13], and GE [14] in endurance sports, and considering the effects of acclimatization for competing at moderate altitudes [15], could be interesting to learn how enhancements in performance in XC skiing are developed. Investigating how altitude interacts with these external and internal variables before and after the competitive season may provide valuable insights into the multifaceted physiological factors affecting performance in junior XC skiers. Altitude exposure should be regarded as a physiological stressor imposed by reduced oxygen availability, eliciting distinct responses depending on the level of exposure. Accordingly, altitude classifications can be outlined as follows: near sea level (0–500 m.a.s.l.) is considered the baseline altitude, with negligible effects on performance; low altitude (500–2000 m.a.s.l.) produces minimal changes, though short-term acclimatization may still be beneficial; moderate altitude (2000–3000 m.a.s.l.) is associated with noticeable reductions in aerobic capacity and an increased risk of acute mountain sickness, requiring longer adaptation periods; high altitude (3000–5500 m.a.s.l.) imposes substantial strain on oxygen transport and performance, even with acclimatization, and prolonged exposure carries significant health risks; above 5500 m.a.s.l., classified as extreme altitude, the human body cannot sustain long-term physiological function, and performance progressively deteriorates [16].

Based on the previously discussed physiological mechanisms induced by training and/or altitude exposure, this study aims to (1) Analyze performance variables in junior XC skiers between normoxic and hypoxic conditions before and after the competitive season. Based on this objective, the hypothesis is based on the expectation that performance variables will differ significantly between normoxic and hypoxic conditions due to the reduction in oxygen availability during exercise; (2) Compare the differences in performance variables under normoxia or hypoxia between pre- and post-training and the competitive season. Based on this objective, the hypothesis is that specific adaptations induced by training and competition under a hypoxic environment will be reflected in the performance variables measured in the study; and (3) Examine the relationship between external load and internal load measured under normoxic and hypoxic conditions, both before and after the training and competitive season, in order to determine whether changes in the measured variables across conditions and time points are related to other study variables and can be predicted through an understanding of the selected variables in the study.

## 2. Material and Methods

### 2.1. Participants

At the start of the study, ten participants were included. Due to an injury, one male participant was excluded, resulting in a final sample of nine participants. In total, seven male and two female XC skiers participated in the study (16.2 ± 1.5 years, 63.6 ± 12.0 kg for males, and 15.8 ± 1.8 years, 59.6 ± 2.5 kg for females). All participants were part of the technical training group of the Navarre Sky Federation (Spain), had a minimum of 2 years of experience in the XC skiing competitions, and participated in national-level races. All participants and their parents gave written consent after being informed of the study procedures.

### 2.2. Procedure

Participants attended the laboratory at two points in the season. The first tests (September and October 2023) were conducted prior to the competitive period (PreCs), during which training routines included ski roller sessions on the road, running, and strength and conditioning sessions. The second tests (March 2024) were conducted at the end of the competitive period, one week after the last competition (i.e., the National Championships race) (PosCs). On each occasion, participants completed two laboratory tests on separate days, one under normoxic and the other under hypoxic conditions, with a 48 h rest period between tests to ensure adequate recovery and consistent testing conditions. Participants were instructed to refrain from vigorous exercise and alcohol ingestion for 48 h prior to testing [17].

The tests were conducted on a motorized treadmill measuring 3 × 4 m. Each session consisted of a Graded Exercise Test (GXT) with 3 min exercise bouts separated by 30 s pauses at the end of each step for capillary blood sample collection for subsequent lactate analysis. The treadmill speed was kept constant at 9 km/h for women and 10 km/h for men, with the slope starting at 1% and increasing by 1% every 3 min [18]. Participants used the classical diagonal stride technique [4] throughout all test stages. Familiarization was achieved with a 5 min warm-up at 8 km/h for both men and women.

Exposure to hypoxic or normoxic conditions was randomized across PreCs and PosCs. Participants were exposed to normoxic or hypoxic air breathing 30 min prior to the start of exercise, ensuring that acute physiological responses under hypoxic conditions had occurred [19]. Participants wore a non-return breathing mask (Hypoxico Biolaster, Biolaster^®^, Andoain, Gipuzkoa, Spain) connected to a normobaric altitude generator (Hypoxico, Hypoxico, Inc.^®^, New York, NY, USA). The normoxic test session was conducted at an equivalent Fraction of Inspired Oxygen (FiO_2_) of 0.209 at 450 m.a.s.l., while the hypoxic conditions simulated an altitude of 2300 m.a.s.l. with a corresponding FiO_2_ = 0.16.

If the estimated partial pressure of oxygen at 1800 m.a.s.l. was considered (130.8 mmHg) without accounting for the water vapor effect at body temperature, the FiO_2_ required to achieve normobaric hypoxia would be 0.17, resulting in a theoretical final Pressure of Inspired Oxygen (PiO_2_) of 130.9 mmHg. However, the effect of body temperature, typically elevated during prolonged high-intensity exercise, must be considered. Applying this theoretical FiO_2_ to reach the hypoxia altitude (final PiO_2_) level during study protocol would not be adequate.

For these reasons, the altitude used in the study protocol was higher than that typically seen in competitions. It was selected based on the comparable difference in PiO_2_ influenced by the water vapor effect at body temperature. Considering the characteristics of the international standard atmosphere, at 1800 m.a.s.l., the total pressure is approximately 623 mmHg. Subtracting the water vapor pressure at 37 °C (47 mmHg), the remaining total pressure is 576 mmHg. From this, the estimated PiO_2_ is 120.9 mmHg (576 × 0.21), representing a difference of 9.9 mmHg from the theoretical previous values. Based on this calculation, the FiO_2_ applied in the study was 0.16. Laboratory conditions were recorded as 22 °C, 50% relative humidity, and approximately 775 mmHg atmospheric pressure, which represented the mean values across the days on which the protocol was conducted. From the atmospheric pressure recorded, the final estimated PiO_2_ applied was 116.5 mmHg ((775 mmHg − 47 mmHg) × 0.16), representing a difference of 4.5 mmHg with estimated corrected values. Finally, considering this PiO_2_ difference (which corresponds to an altitude approximately 260 m higher than 1800 m.a.s.l.), as well as the short duration of exercise performed by participants and the lower likelihood of reaching a body temperature peak comparable to that of competition or intense training, this PiO_2_ was ultimately selected for the hypoxia exposure protocol.

Participants were blinded to the experimental condition, helping to eliminate biases and ensure that any differences in their performance were not influenced by their knowledge of the environment.

HR was continuously recorded using a Polar H10 monitor (Polar Electro Oy®, Kempele, Finland) and averaged over the final 30 s of each step. Blood lactate accumulation was measured by collecting capillary blood samples from the earlobe during the 30 s rest period between steps and analyzed using a Lactate Pro 2 analyzer (Lactate Pro, Arkray, KDK Corporation^®^, Kyoto, Japan) [20].

### 2.3. Data Analysis

Heart rate is presented as beats per minute (bpm), and slope is presented as the percentage of treadmill inclination (%). Both parameters were calculated at a lactate concentration of baseline lactate plus an increment of 1 mmol/L (Bsln+1.0) [21] and at a fixed concentration of 4 mmol/L (OBLA4) [5,22]. A third-order polynomial curve was used to calculate these points. Bsln+1.0 and OBLA4 were calculated using the “Lactater” package (Mattioni Maturana F (2023)—Lactater: Tools for Analyzing Lactate Thresholds; R package version 0.2.0).

### 2.4. Statistical Analysis

Values are expressed as mean ± standard deviation (SD). Normality was assessed using the Shapiro–Wilk test and subsequently re-evaluated with a Quantile–Quantile plot. Differences between normoxia and hypoxia test conditions within PreCs or PosCs were examined using a paired T-test. Differences between PreCs and PosCs tests under normoxia or hypoxia were initially considered for analysis of variance (ANOVA), but due to the small sample size and normality of results, Wilcoxon’s test was used. Correlation analysis was performed using Pearson’s coefficient of correlation (r). Regression analysis is performed based on linear regression, presenting probability and a coefficient of determination (R^2^).

The probability values from the statistical tests performed (p), the mean difference expressed as a percentage (difference), and the effect sizes (d) are provided, with the latter classified as small (d  ≥ 0.2 < 0.5), medium (d  ≥  0.5 < 0.8) and large (d ≥ 0.8) [23]. Confidence intervals (CI) at 95% were provided for variables where variability estimation was relevant, following the application of the appropriate statistical tests.

Based on previous studies, the main effect of altitude exposure on performance variables at 2000 m.a.s.l is approximately a 15–15.6% decrease from sea level values [24,25]. Following this, an effect size of 1.1 (Cohen’s d) [23] was set as a minimum detectable. The statistical power of the study was set at 80%, with alpha levels at 0.05. Considering the same group intervention in both hypoxia and normoxia tests, the mean fixed value was 0, with the means and standard deviations of Group 1 and Group 2 being considered equal. Using the group characteristics and statistical requirements mentioned earlier, the estimated sample size was 9 participants, with a non-centrality parameter of 3.3, a Type I error rate of 0.05, and a Type II error rate of 0.2.

All statistical analyses were performed using R Studio (RStudio Team, 2020; RStudio: Integrated Development for R. RStudio, PBC, Boston, MA, USA. URL http://www.rstudio.com/, accessed on 12 October 2021). The statistical significance level was set at *p* ≤ 0.05.

## 3. Results

The maximum values of lactate, heart rate, and slope reached during the GXT under normoxia and hypoxia for PreCs and PosCs are presented in Table 1. There were no significant differences between normoxia and hypoxia in PreCs or PosCs for maximum lactate, heart rate, and slope. Conversely, significant differences were observed in maximum lactate and heart rate under hypoxia between PreCs and PosCs, as well as in maximum lactate under normoxia between PreCs and PosCs.

Results of lactate, heart rate, and slope estimated at Bsln+1.0 and OBLA4 under normoxia and hypoxia at PreCs and PosCs are presented in Table 2. The slope estimated under hypoxia at Bsln+1.0 and OBLA4 was statistically different from normoxia values at PreCs. At PosCs, the mean slope at OBLA4 was significantly different between conditions, but for Bsln+1.0, no significant effects were found. The Wilcoxon’s test results did not show significant changes between PreCs and PosCs variables under both hypoxia and normoxia. Lactate at Bsln+1.0 showed no differences between conditions or timepoints, as well as HR at both Bsln+1.0 and OBLA4.

Correlation analysis during the GXT under normoxia and hypoxia in PreCs and PosCs is shown in Figure 1. Significant correlations were found under hypoxia at PreCs between heart rate at OBLA4 and heart rate at Bsln+1.0, with a very large effect (*p* = 0.002, 95% CI [0.5, 1.0], r = 0.88), and a very strong correlation between slope at OBLA4 and slope at Bsln+1.0 (*p* = 0.003, 95% CI [0.4, 1.0], r = 0.86). Under normoxia, very strong correlations were found between heart rate at OBLA4 and heart rate at Bsln+1.0, and between slope at OBLA4 and slope at Bsln+1.0 (*p* < 0.001, 95% CI [0.8, 1.0], r = 0.97, and *p* < 0.001, 95% CI [0.8, 1.0], r = 0.97, respectively).

Under hypoxia at PosCs, a negative, statistically significant, and very large correlation was found between heart rate at OBLA4 and lactate at Bsln+1.0 (*p* = 0.028, 95% CI [−0.9, −0.1], r = −0.72). Additionally, correlations between heart rate at OBLA4 and heart rate at Bsln+1.0, as well as between slope at OBLA4 and slope at Bsln+1.0, were identified in PosCs, with a very large effect (*p* < 0.001, 95% CI [0.7, 1.0], r = 0.93, and *p* = 0.003, 95% CI [0.4, 1.0], r = 0.86, respectively). Under normoxia, a very large correlation between heart rate at OBLA4 and Bsln+1.0 was still present (*p* < 0.001, 95% CI [0.9, 1.0], r = 0.98), along with a very large effect between slope at OBLA4 and Bsln+1.0 (*p* < 0.001, 95% CI [0.9, 1.0], r = 0.97). Moreover, Slope at Bsln+1.0 was very largely and positively correlated with heart rate at Bsln+1.0 (*p* = 0.043, 95% CI [0.0, 1.0], r = 0.68), and slope at OBLA4 was also very largely and positively correlated with heart rate at Bsln+1.0 (*p* = 0.046, 95% CI [0.0, 1.0], r = 0.68).

Delta of slope at Bsln+1.0 and OBLA4, calculated as the difference in absolute values between hypoxia and normoxia within PreCs or PosCs, are presented in Table 3. Analyzing differences in Delta values between PreCs and PosCs, a significant statistic difference was found in Slope at Bsln+1.0.

At PreCs, neither Delta values in slope at Bsln+1.0 with slope at Bsln+1.0 at normoxia nor Delta values in slope at OBLA4 with slope at OBLA4 at normoxia were significantly correlated (*p* = 0.819, 95% CI [−0.7, 0.6], r = −0.09 and *p* = 0.593, 95% CI [−0.8, 0.5], r = −0.21, respectively).

At PosCs, Delta values at Bsln+1.0 were correlated significantly with slope at Bsln+1.0 at normoxia (*p* = 0.048, 95% CI [−0.9, 0.0], r = −0.67) and Delta values between hypoxia and normoxia slope estimated at OBLA4 and OBLA4 at normoxia (*p* = 0.031, 95% CI [−0.9, 0.0], r = −0.71).

### Regression Analysis

At PosCs, Slope at Bsln+1.0 under normoxia explains 37% of the variance in Delta of slope at Bsln+1.0 between normoxia and hypoxia (*p* = 0.049, R^2^ = 0.45, adjusted R^2^ = 0.37). In addition, 78% of variance of Delta in slope at OBLA4 between normoxia and hypoxia was explained by slope reached at OBLA4 under normoxia (*p* < 0.001, R^2^ = 0.51, adjusted R^2^ = 0.78).

## 4. Discussion

The purpose of this study was that performance variables would differ significantly between normoxic and hypoxic conditions due to reduced oxygen availability during exercise before and after the training and competitive season. Results did not show significant differences in maximal lactate concentration, heart rate, and slope between normoxia and hypoxia at both PreCs and PosCs, indicating that while hypoxia may influence physiological responses, the maximal performance variables did not differ markedly under these conditions. On the other hand, slope at Bsln+1.0 and OBLA4 showed significant differences between normoxia and hypoxia at PreCs, with the slope reached at OBLA4 at PosCs also exhibiting a significant decrease. This indicates a negative effect of hypoxic exposure on performance, observed across different time points measured throughout the study. These findings could be due to the physiological effects of the training and competitive period, particularly in variables such as lactate and heart rate, which may have been influenced by the demands of the training and competition period.

The next commented hypothesis in the study was that possible differences in internal and external load variables, when comparing PreCs and PosCs within the same condition test, could be found. In contrast, no significant differences were observed in the other variables when comparing PreCs and PosCs within the same conditions.

The last objective and hypothesis, which posited that changes across conditions and time points would correlate with other study variables and could be predicted through their understanding, the correlation analysis revealed stronger relationships between heart rate at Bsln+1.0 and heart rate at OBLA4 under normoxia, with a strong correlation observed at PreCs, which became even stronger at PosCs in both conditions, especially under hypoxia. Finally, significant correlations were found between the Delta values (between hypoxia and normoxia) for slope at Bsln+1.0 and OBLA4, with the slope reached under normoxic conditions at each variable. These correlations, along with the regression analysis, highlight the relationship between responses to specific exercise environments and performance measured under normoxic conditions, which are considered as a “basal, non-altered environment” performance.

### 4.1. Maximal Values During the GXT

The findings from this study reveal no significant differences between hypoxia and normoxia at either PreCs or PosCs regarding maximum values of lactate, heart rate, and slope. In the case of slope, considered as external load, it aligns with previous findings, which indicate that while the fractional utilization of oxygen consumption for achieving an external load in runners is influenced by hypoxia (FiO_2_ = 0.18), the external load achieved by athletes (i.e., race pace) is not modified in a maximal test [26]. There was no significant difference in maximal heart rate between hypoxia and normoxia in PreCs and PosCs, contradicting previous findings that acute exposure to hypoxic exercise limits maximum heart rate by 1.3–1.6 beats per minute for every 1000 m of altitude [12,27]. Lactate values were not significantly altered, indicating a comparable physiological response in lactate values measured during a maximal test under normoxic and hypoxic conditions [12,27].

Comparing assessed variables before and after the training and competitive season under normoxia or hypoxia, no significant differences were found between the maximal values in heart rate and slope reached in a GXT under normoxic conditions. However, significant changes in maximum lactate under normoxia were observed between PreCs and PosCs. Additionally, under hypoxic conditions, significant changes were found between PreCs and PosCs in values of maximum lactate and heart rate. Despite our hypothesis having been partially confirmed, the changes in the commented variables were lower in PosCs, showing a negative effect after the training and competition period on the maximum values reached in the lactate and heart rate condition under hypoxia, as well as lactate under normoxia. These results could be attributable to a reduced capacity to reach maximum heart rate as a limited central factor and a shifted metabolic contribution during exercise, as indicated by maximum lactate values [28].

Significant reductions in commented variables were found after the training and competitive periods, which consisted of an average of 100 h of training at altitudes ranging from 1800 to 2300 m above sea level, as well as participation in an average of 14 competitions conducted within the same altitude range, under both normoxic and hypoxic conditions. Given this potential detrimental effect, the concept of the fitness-fatigue state (by the load achieved by participants in study) must be acknowledged as a possible confounding factor in assessing performance changes throughout the season in cross-country skiers [29]. Also, the difference in performance could be due to the athlete’s previous experiences in a hypoxic state. According to the central governor theory, previous experiences can, to some extent, determine the onset of fatigue without there really being a peripheral fatigue [30], likewise, Amann et al. (2006) [31] demonstrated that 5 km cycling time trial performance was impaired with hypoxia and improved with hyperoxia, relative to normoxia, yet peripheral (quadriceps) fatigue was not different between the conditions. This was interpreted as indicating that the locomotor muscle output is determined to a significant extent by the regulation of central motor output to the working muscle, in order that peripheral muscle fatigue does not exceed a critical threshold [31].

In addition, the psychophysiological condition of these young participants could be influenced by both physiological and psychological factors [32] following a strenuous period of training and competition. All these possible situations that may lead to differences in performance enhancements should be considered in future studies when conducting a GXT, following a training and competition period, on young XC skiers.

### 4.2. Slope Values at Bsln+1.0 and OBLA4

Differences in the slope reached at Bsln+1.0 and OBLA4 between hypoxia and normoxia at PreCs suggest a reduction in the submaximal ability to perform under hypoxic conditions at the first and second lactate thresholds detected during a GXT. These changes could be attributed to the altered metabolic demands imposed by hypoxic conditions during exercise [12,33]. Specifically, the significant difference in slope at OBLA4 may indicate a modification in the aerobic–anaerobic balance mediated by hypoxic exposure and the close relationship between the external load of OBLA4 and the Maximum Lactate Steady State (MLSS) in XC skiing [22].

Interestingly, at PosCs, comparing hypoxia and normoxia slope values at Bsln+1.0, results were not statistically different, showing a comparable external load achieved at Bsln+1.0 under both conditions, possibly due the effect of training and competitive periods, which allowed skiers to perform under specific hypoxic conditions without an “impaired effect” of hypoxia on low-intensity external load performance. In addition, despite slope values between normoxia and hypoxia at PosCs for OBLA4 being statistically lower in the latter, the mean difference and Cohen’s d value between normoxia and hypoxia at PreCs and PosCs differed, showing a “less impairment effect” in slope decrease during hypoxia GXTs compared to normoxia at PosCs: from a mean difference = −0.85 in PreCs to d = −0.44 in PosCs. These changes could be explained by an “acclimatization” effect to hypoxia, as athletes trained and competed at altitudes between PreCs and PosCs. In fact, an increase of 0.9 km/h (7%) in external load at the first ventilatory threshold generated under hypoxic conditions after a training period under hypoxia has been shown before in cycling and running athletes [34,35]. Despite the significant results suggesting a possible “acclimatization” effect, these findings should be interpreted with caution due to the limited scope of the altitude exposure, which included only hypoxia and did not consider other additive factors such as cold. Cold exposure has been shown to affect lactate thresholds and workload when combined with hypoxia, compared to hypoxia alone, particularly in running exercise [36]. In this way, it is important to consider the potential synergistic effect, which could add to the impact of hypoxic exposure due to altitude in XC ski competitions. Future studies should also examine the potential impact of other environmental factors (such as cold, wind, radiation, etc.) on performance during altitude exposure.

Similar PreCs and PosCs values of slope at Bsln+1.0 and OBLA4 under normoxia or hypoxia align with non-significant improvements in VO_2_ at the second ventilatory threshold in Nordic skiers after the training and competitive season [37]. This could explain the lack of enhancement between PreCs and PosCs slope variables in this study, especially for OBLA4 values of slope, which are strictly related to aerobic–anaerobic contribution during exercise and the possible fatigue state after the training and competitive periods. Considering this possible detrimental effect of fatigue, the concept of a fitness-fatigue state in athletes must be recognized as a potential confounding factor in evaluating performance changes over the course of the season in cross-country skiers [29]. Therefore, our hypothesis has been partially confirmed.

### 4.3. Correlations Between External and Internal Variables

At PreCs under normoxia, correlations between heart rate at Bsln+1.0 and heart rate at OBLA4 and between slope at Bsln+1.0 and slope at OBLA4 were found, all with an r value of 0.97. In case of hypoxia, a decreased relationship effect was found, with a r = 0.88 between heart rate at Bsln+1.0 and heart rate at OBLA4, and r = 0.86 between slope at Bsln+1.0 and slope at OBLA4. These results suggest that relationships between GXT variables under normoxia are stronger compared to those under hypoxia at PreCs.

After the training and competitive periods, Pearson’s r coefficient between heart rate and slope at Bsln+1.0 and OBLA4 under normoxia remained close to 0.98. Interestingly, the r value under hypoxia between heart rate at Bsln+1.0 and heart rate at OBLA4 increased to 0.93, while the correlation between slope at Bsln+1.0 and slope at OBLA4 remained stable at PreCs values (i.e., r = 0.86).

Comparing the degree of relationships between variables under hypoxia, an increase in Pearson’s correlation coefficient between heart rate at Bsln+1.0 and heart rate at OBLA4 was detected from PreCs to PosCs. This suggests that the relationship between this internal load variable before a training and competitive period in XC skiing during a GXT has a certain degree of correlation, but after the training and competitive periods, the strength of this correlation is higher. Correlation of threshold’s variables results has been found before under hypoxic or normoxic conditions [38], but following these results, the strength of relationships between GXT variables under hypoxia increases after a training and competitive period, showing a more stable determination of internal load variables from lactate thresholds of Bsln+1.0 and OBLA4. Therefore, our hypothesis has been confirmed by observing an increase in the strength of the relationship between the internal load variables estimated at different thresholds in a GXT under hypoxia after a period of training and competition in a hypoxic environment.

### 4.4. Effect of Hypoxia Under External Load Threshold Variables

Delta values in the variables of slope at Bsln+1.0 and OBLA4 suggest a considerable impairment effect of hypoxic exposure during a GXT. The values, ranging from −0.43 to −1.25, represent a relative percentage value of the absolute load to be developed, corresponding to a range from 3.5 to 10% performance loss in external load. These findings are consistent with a 6.3% decrease in performance for every 1000 m of altitude, as previously observed in endurance [12]. Despite the impairment effect of hypoxia, Delta changes between hypoxia and normoxia in slope at Bsln+1.0 showed a decrease after training and competitive periods, switching from −1.25 to −0.44. Similarly, OBLA4 values also shifted, from −0.85 to −0.43. Both changes could be considered as a possible adaptation effect attributed to training and competing in specific environments. Although this potential adaptation must be considered with caution and needs confirmation by future studies, correlation analysis using Delta changes in slope estimated at different thresholds between hypoxia and normoxia at PosCS reinforces the statistical relationship between the level of performance of skiers reached in a GXT under normoxia and the impairment effect of hypoxia under performance. This finding is consistent with previous studies that have examined the relationship between skier performance levels and performance loss (VO_2_max) under hypoxic conditions [38]. In addition, significant correlations found after the training and competitive periods could help explain the specific changes in performance variables under environmental conditions of reduced PiO_2_ under a GXT after a training and competition period in XC skiing.

Finally, a comprehensive regression analysis of the Delta change in external load performance at various thresholds, with external load estimated at these points, provides a novel approach to understanding XC ski performance and the possible related factors of success within the sport and the intrinsic environmental conditions. This analysis highlights how performance is influenced by environmental conditions and is further explained by the skill level of the skiers. These findings are novel and should be interpreted with caution. Further studies are necessary to validate these results and confirm their reproducibility.

### 4.5. Strengths, Limitations, Practical Applications, and Future Directions

This study offers a detailed evaluation of physiological and performance responses to normoxia and hypoxia in young cross-country skiers before and after a training and competitive season. The combined use of internal and external load measures during a GXT under hypoxic and normoxic conditions, along with correlation and novel regression analyses, provides valuable insights into acclimatization effects and environmental influences on performance following a specific period of training and competition in a hypoxic environment.

Despite these results, several limitations of the study must be acknowledged. First, the sample size calculation was based on statistical power derived from the expected effect of an external load variable, for which the most reliable prior estimates were available. However, since statistical power is specific to each endpoint, this calculation may not have fully extended to the internal load outcomes, such as mean heart rate and lactate responses. These internal load variables are often subject to greater within and between subject variability, and the true effects may be smaller due to individual physiological differences. As a result, the study may have been underpowered to detect certain effects in the internal load metrics of heart rate and lactate. To facilitate interpretation, effect sizes with 95% confidence intervals are reported for all internal load outcomes, and these results should be considered exploratory. Another limitation is the lack of detailed descriptive training data from both the coaches and the athletes themselves. Comprehensive information on the athletes’ individual training loads and periodization would have provided a more complete understanding of the internal and external load responses observed during the study. Additionally, the limited hypoxic exposure (without additional stressors such as cold) restricts the generalizability of the findings. Furthermore, potential fatigue following the competitive season must be considered when assessing young XC skiers in a GXT after a period of training and competitive stress. The results suggest that altitude competition and training periods may reduce hypoxia-related performance changes in a GXT. Monitoring both internal and external loads can help to better understand performance changes under specific conditions and the differences in performance before and after training and competition periods at altitude.

Larger, more diverse samples and inclusion of combined environmental stressors are needed. Coaches and sport scientists can use these findings as a starting point to better understand how hypoxia affects submaximal and maximal performance variables, how acclimatization over a training season can mitigate hypoxia-related responses in exercise, and how specific environmental competition conditions influence performance. Future studies should assess psychophysiological aspects, refine acclimatization protocols, and investigate underlying mechanisms of hypoxia adaptation in XC skiers after a training and competitive season.

## 5. Conclusions

This study investigated the physiological responses of cross-country skiers before and after the competitive season, as well as the potential impact of normoxic and hypoxic conditions on their performance capacity.

The results revealed comparable maximum lactate, heart rate, and slope values between normoxia and hypoxia measured in a GXT before and after the training and competitive period, suggesting a consistent physiological response during maximal exertion under both conditions. Despite this, some differences between hypoxia and normoxia in slope reached at different thresholds were found in PreCS. However, after the training and competitive period, slope at Bsln+1.0 was not statistically different, showing a possible acclimatization effect of training and competitive period on this variable. Correlation analysis between internal (heart rate) and external (slope) loads at estimated thresholds showed stronger correlations after the competitive season, particularly under hypoxic conditions.

Considering the relationship between skiers’ ability to develop external load (slope reached at different thresholds under normoxic conditions) and the impairment effects (Delta) observed between hypoxic and normoxic conditions, regression analysis indicated a potential causal relationship. This indicates that the level of external load achieved by skiers at different thresholds under normoxia can predict the impairment effects under hypoxia compared to normoxia in a GXT.

Environmental factors, training, and competition affect XC junior skiers’ performance, emphasizing the need to better understand hypoxia’s role. Replicating hypoxic GXTs could help evaluate seasonal performance changes across skier categories. However, individual variability and training specifics may influence results, requiring further research to validate and expand these findings.

## Figures and Tables

**Figure 1 sports-13-00344-f001:**
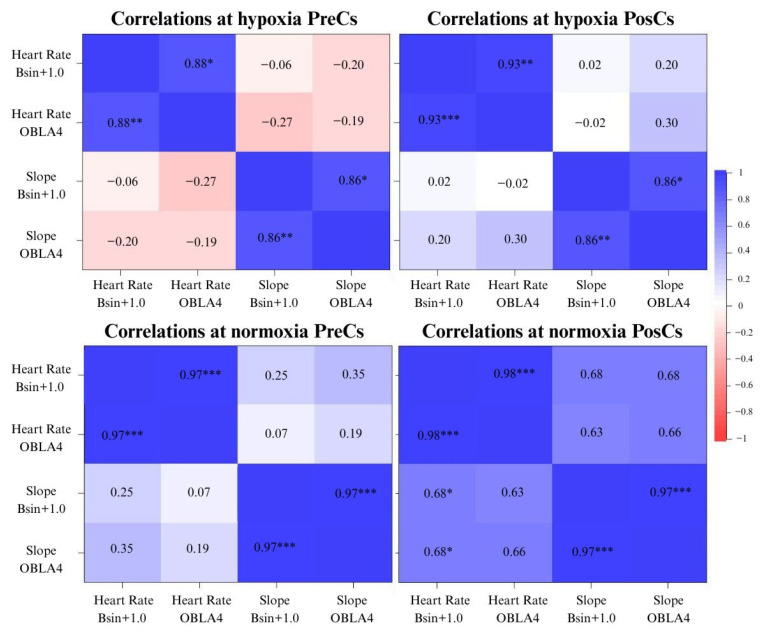
Correlation plots between variables of lactate, heart rate, and slope under normoxia and hypoxia at PreCs and PosCs. Note: Lactate Bsln+1.0: lactate estimated at the value of baseline lactate plus 1 mmol/L during the GXT. Heart Rate Bsln+1.0: heart rate estimated at the value of baseline lactate plus 1 mmol/L during the GXT. Heart Rate OBLA4: heart rate estimated at the lactate value of 4 mmol/L during the GXT. Slope Bsln+1.0: slope estimated at the value of baseline lactate plus 1 mmol/L during the GXT. Slope OBLA4: slope estimated at the lactate value of 4 mmol/L during the GXT. * *p* < 0.05, ** *p* < 0.01, *** *p* < 0.001.

**Table 1 sports-13-00344-t001:** Maximum results of lactate, heart rate, and slope at the GXT under hypoxia and normoxia before and after the competitive season.

Variable		PreCS	PosCs	Result Between PreCs and PosCs	Normoxia vs. Hypoxia in PreCs	Normoxia vs. Hypoxia in PosCs
		Mean ± Standard Deviation	Mean ± Standard Deviation	Wilcoxon’s Test Values	*p* d	*p* d
HYPOXIA	Maximum Lactate (mmol/L)	8.6 ± 2.7	7.1 ± 2.2 #	*p* = 0.011, difference = 1.5, 95% CI [0.6, 2.3], d = 1.1, 95% CI [−0.5, 2.8], large	*p* = 0.144, difference = 0.9, 95% CI [−0.4, 2.1], d = 0.5, 95% CI [−0.2, 1.2], medium	*p* = 0.803, difference = 0.2, 95% CI [−1.6, 2.0], d = 0.1, 95% CI [−0.6, 0.7]
	Maximum Heart Rate (bpm)	195 ± 8	191 ± 9 #	*p* = 0.012, difference = 4, 95% CI [2, 6], d = 1, 95% CI [−1, 3], large	*p* = 0.657, difference = 1, 95% CI [−2, 3], d = 0, 95% CI [−1, 1]	*p* = 0.427, difference = −1, 95% CI [−4, 2], d = 0, 95% CI [−1, 0]
	Maximum Slope (%)	10 ± 2.0	9.2 ± 1.3	*p* = 0.057, difference = 0.9, 95% CI [0.3, 1.5], d = 1, 95% CI [−0.5, 2.0], large	*p* = 0.896, difference = 0, 95% CI [−0.7, 0.8], d = 0, 95% CI [−0.6, 0.7]	*p* > 0.999, difference = 0, 95% CI [−0.7, 0.7], d = 0, 95% CI [−0.6, 0.6]
NORMOXIA	Maximum Lactate (mmol/L)	7.7 ± 2.2	6.9 ± 3.4 #	*p* = 0.028, difference = 0.8, 95% CI [−0.5, 2.1], d = 0.4, 95% CI [−0.2, 1.6], small		
	Maximum Heart Rate (bpm)	195 ± 9	192 ± 9	*p* = 0.105, difference = 2, 95% CI [0, 5], d = 1, 95% CI [0, 2], large		
	Maximum Slope (%)	10 ± 1.8	9.2 ± 1.7	*p* = 0.091, difference = 0.9, 95% CI [0.2, 1.5], d = 0.9, 95% CI [−0.2, 2.9], large		

bpm: beats per minute. CI: confidence interval. d: Effect size (Cohen’s d). *p*: probabilistic values of statistic test performed. **%**: percentage of slope. #: significative difference between same test condition (hypoxia or normoxia) between PreCs and PosCs.

**Table 2 sports-13-00344-t002:** Results of lactate, heart rate, and slope, at estimated values of Bsln+1.0 and OBLA 4 mmol from the GXT under hypoxia and normoxia before and after the competitive season.

Variable		PreCS	PosCs	Result Between PreCs and PosCs	Normoxia vs. Hypoxia in PreCs	Normoxia vs. Hypoxia in PosCs
		Mean ± Standard Deviation	Mean ± Standard Deviation	Wilcoxon’s Test Values	*p* d	*p* d
HYPOXIA Bsln+1.0	Lactate (mmol/L)	2.5 ± 0.2	2.5 ± 0.2	*p* = 0.550, difference = 0, 95% CI [−0.1, 0.1], d = 0.2, 95% CI [−1.0, 0.8], small	*p* = 0.053, difference = 0.10, 95% CI [−1.7 × 10^−3^, 0.2], d = 0.8, 95% CI [−9.6 × 10^−3^, 1.5], large	*p* = 0.111, difference = 0.1, 95% CI [0, 0.3], d = 0.6, 95% CI [−0.1, 1.3], medium
	Heart Rate (bpm)	159 ± 15	164 ± 10	*p* = 0.406, difference = 5, 95% CI [−4, 13], d = 0, 95% CI [0, 1]	*p* = 0.197, difference = −6, 95% CI [−15, 4], d = 0, 95% CI [−1, 0]	*p* = 0.325, difference = −3, 95% CI [−10, 4], d = 0, 95% CI [−1, 0]
	Slope (%)	4.5 ± 2.0 *	5.1 ± 1.2	*p* = 0.250, difference = 0.6, 95% CI [−1.0, 2.3], d = 0.2, 95% CI [−0.3, 1.4], small	*p* = 0.005, difference = −1.3, 95% CI [−2.0, −0.5], d = −1.3, 95% CI [−2.2, −0.3], large	*p* = 0.165, difference = −0.4, 95% CI [−1.1, 0.2], d = −0.5, 95% CI [−1.2, 0.2], medium
HYPOXIA OBLA4	Heart Rate (bpm)	178 ± 16	179 ± 12	*p* = 0.440, difference = 1, 95% CI [−5, 6], d = 0, 95% CI [0, 2]	*p* = 0.111, difference = −4, 95% CI [−10, 1], d = −1, 95% CI [−1, 0], large	*p* = 0.056, difference = −4, 95% CI [−8, 0], d = −1, 95% CI [−1, 0], large
	Slope (%)	6.7 ± 1.8 *	6.9 ± 1.3 *	*p* = 0.496, difference = 0.1, 95% CI [−0.3, 1.6], d = 0.1, 95% CI [−0.5, 1.3]	*p* = 0.013, difference = −0.8, 95% CI [−1.5, −0.2], d = −1.1, 95% Cl [−1.9, −0.2], large	*p*= 0.02, difference = −0.4, 95% CI [−0.8, −0.1], d= −0.9, 95% CI [−1.7, −0.1], large
NORMOXIA Bsln+1.0	Lactate (mmol/L)	2.4 ± 0.2	2.4 ± 0.2	*p* = 0.714, difference = 0, 95% CI [−0.1, 0.1], d = −2.7, 95% CI [−1.2, 0.5], large		
	Heart Rate (bpm)	165 ± 16	167 ± 17	*p* = 0.341, difference = 2, 95% CI [−6, 10], d = 0, 95% CI [0, 2]		
	Slope (%)	5.7 ±1.8	5.5 ± 1.6	*p* = 0.652, difference = −0.2, 95% CI [−1.8, 1.4], d = −0.1, 95% CI [−0.6, 1.2]		
NORMOXIA OBLA4	Heart Rate (bpm)	183 ± 19	183 ± 16	*p* = 0.635, difference = 0, 95% CI [−5, 6], d = 0, 95% CI [0, 1]		
	Slope (%)	7.6 ± 1.7	7.3 ± 1.6	*p* = 0.734, difference = −0.3, 95% CI [−1.7, 1.1], d = −0.1, 95% CI [−0.6, 1.0]		

bpm: beats per minute. CI: confidence interval. d: Effect size of Cohen’s d. *p*: probabilistic values of statistic test performed. %: percentage of slope. *: significative difference between hypoxia and normoxia test within same time of measurement.

**Table 3 sports-13-00344-t003:** Results of Delta values between hypoxia and normoxia before and after the competitive season at estimated Slope of Bsln+1.0 and OBLA 4 mmol from the GXT.

Variable	∆ Between Hypoxia vs. Normoxia PreCs	∆ Between Hypoxia vs. Normoxia PosCs	Wilcoxon Test Results Between ∆ PreCs vs. PosCs
Slope at Bsln+1.0 (%)	−1.2 ± 1.0 *	−0.4 ± 0.8	*p* = 0.020; 95% CI [−1.0, −0.5]
Slope at OBLA 4 (%)	−0.8 ± 0.8	−0.4 ± 0.4	*p* = 0.098; 95% CI [−0.9, 0.0]

∆: Delta value. *p*: *p* values. *: significative difference between PreCS vs. PosCs within condition: hypoxia or normoxia.

## Data Availability

The data presented in this study are available on request from the corresponding author due to (data were provided under a data-sharing agreement).

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
