# Peer review of "Lactate Thresholds and Performance in Young Cross-Country Skiers Before and After the Competitive Season: Insights from Laboratory Roller-Ski Tests in Normoxic and Hypoxic Conditions"

_sports, 2025, doi:10.3390/sports13100344_

Round 1
Reviewer 1 Report
Comments and Suggestions for Authors
Dear Authors,
I congratulate you for this interesting study, which would be of interest to the readers and which contributes to the knowledge in the field. However, I have the following points to raise:
- In the introduction you mention that XC competitions are often conducted between sea level and 1800 m.a.s.l. I would therefore ask the authors to a) explain why a simulated altitude of 2300 m.a.s.l. was chosen for the protocol and b) to discuss the overall relevance of the results when considering the majority of XC competitions take place at much lower altitudes. Especially, an overview would be desirable that states the amount or percentage of XC races taking place at high altitudes, e.g. above 1500 m.a.s.l. In that regard, a definition of the various altitudes and their physiological impact should briefly be introduced and discussed.
- The number of nine participants is comparatively low. Yet, the authors clearly describe that a statistical power of 80% was aimed at. Becuadse of the several non-significant effects, I would like to ask the authors if a power of 80% was achieved in all these cases.
- The authors described the cimcumstances of the testing taking place under controlled conditions. However, they would appear to be quite different regarding environmental conditions during the actual XC competitions, e.g. a XC race in cold environment, with wind chill effects, et cetera. I would therefore like to ask the authors to a) more clearly state these conditions during the testing, such as temperature, humidity, wind, et cetera, and b) to more thoroughly discuss the impact of possible differences of that conditions to actual XC competitions.
- The tables are a biut cumbersome to read (format and lots of information in certain cells), perhaps there is room for improvement.
- Figure 1 shows the mentioned correlations, however, 50% of the data shown is superfluous, since the correlation between for example Slope OBLA4 and Heart Rate OBLA4 is the same as between Heart Rate OBLA4 and Slope BBLA4. Furthermore, it should be clear that each parameter shows a correlation coefficient of 1 with itself, so I would suggest to delete these data.
- In line 298 that authors mention that the test subjects trained and competed at altitudes between PreCs and PosCs. Are there more deatiled descriptions regarding the altitude and the durantion during which the subject were training and competing under these conditions?
- In general, with regards to fatigue, I would suggest to the authors to consider and discuss also mental/psychological causes for fatigue (e.g. central governor theory), especially when discussing the changes from the pre to post competion phases.
Many kind regards!
Comments on the Quality of English LanguageDear Authors,
I suggest to thoroughly check orthography. I detected several typos, including missing or superfluous spaces, e.g. lines 24, 74, 244, 290, 377.
Many kind regards!
Author Response
To Reviewers in response to manuscript review Round 1:
Dear Reviewer, 1:
Thank you very much for your insightful comments and feedback on our study. We truly appreciate your recognition of the value of our work, and we are grateful for the opportunity to improve the manuscript.
In response to your points, we have made the necessary revisions and incorporated the changes accordingly. Below, we outline the modifications we have made to the manuscript for your review.
- In the introduction you mention that XC competitions are often conducted between sea level and 1800 m.a.s.l. I would therefore ask the authors to a) explain why a simulated altitude of 2300 m.a.s.l. was chosen for the protocol and b) to discuss the overall relevance of the results when considering the majority of XC competitions take place at much lower altitudes. Especially, an overview would be desirable that states the amount or percentage of XC races taking place at high altitudes, e.g. above 1500 m.a.s.l. In that regard, a definition of the various altitudes and their physiological impact should briefly be introduced and discussed.
Response 1: a) We We have accordingly revised and included the rationale for selecting a higher altitude for hypoxic load exposure (FiOâ‚‚) based on the difference between the theoretical PiOâ‚‚ at a chosen altitude and the effect of body temperature under water vapour pressure, which displaces the total pressure of dry air. This ultimately determines the PiOâ‚‚ available for the respiratory system.
We believe that the PiOâ‚‚ used in the hypoxia exposure protocol in the study closely reflects the actual PiOâ‚‚ experienced by XC skiers in real competitions. Due to notable technical limitations in presenting studies conducted in real competition environments, we support the PiOâ‚‚ used in this study based on previously discussed evidence relating to normobaric and hypobaric hypoxia, as well as the differences in PiOâ‚‚ that contribute to the genuine constraints of employing this type of hypoxia.
If only the partial pressure of oxygen at 1800 m a.s.l. were considered without adjusting for water vapour at 37 °C, the resulting FiOâ‚‚ would underestimate the actual hypoxic stress. At body temperature, the water vapour pressure (47 mmHg) reduces the available pressure for oxygen, thereby lowering the true Pioâ‚‚. To more accurately reflect this physiological condition, a FiOâ‚‚ of 0.16 was applied under controlled laboratory conditions (22 °C, 50% relative humidity, 775 mmHg), resulting in an estimated PiOâ‚‚ of 116.5 mmHg. This represents a 4.5 mmHg difference compared to the corrected PiOâ‚‚ at 1800 m a.s.l., which corresponds to an equivalent altitude of approximately 2060 m.
This adjustment was also justified by the short duration of the exercise protocol, during which participants were unlikely to reach core temperatures comparable to those experienced during competition or intense training (close to ~39 °C), at which the PiOâ‚‚ would be approximately 2 mmHg higher than the PiOâ‚‚ applied in this study. Therefore, the selected PiOâ‚‚ was considered more appropriate to induce a comparable level of normobaric hypoxia in the laboratory setting.
These changes are reflected on the 3rd page, 5th paragraph, starting from line 129.
- b) Since we consider the PiOâ‚‚ applied in the hypoxic protocol to be equivalent to that in real XC ski competitions, we have not discussed the differences between the altitude at competitions and the theoretical altitude used in the study, with normobaric hypoxia exposure.
Girard, O., Koehle, M. S., MacInnis, M. J., Guenette, J. A., Koehle, M. S., Verges, S., Rupp, T., Jubeau, M., Perrey, S., Millet, G. Y., Chapman, R. F., Levine, B. D., Conkin, J., Wessel, J. H., 3rd, Nespoulet, H., Wuyam, B., Tamisier, R., Verges, S., Levy, P., Casey, D. P., … Nelatury, C. F. (2012). Comments on Point:Counterpoint: Hypobaric hypoxia induces/does not induce different responses from normobaric hypoxia. Journal of applied physiology (Bethesda, Md.:1985), 112(10),1788–1794. https://doi.org/10.1152/japplphysiol.00356.2012
A definition of the various altitudes and their physiological impacts has been added and discussed in the introduction. These changes are reflected on the 2nd page, 5th paragraph, starting from line 88.
- The number of nine participants is comparatively low. Yet, the authors clearly describe that a statistical power of 80% was aimed at. Becuadse of the several non-significant effects, I would like to ask the authors if a power of 80% was achieved in all these cases.
Response 2: Our a priori sample size calculation was based on the expected mean effect in an external load variable, for which we had the most robust prior estimates. As statistical power is specific to each endpoint (variable), this calculation may not fully extend to internal load outcomes (in our case, mean Heart Rate and Lactate responses), which often exhibit greater within and between subject variability and may present smaller true effects due to individual physiological differences. Consequently, the study may have been underpowered to detect certain effects in some internal load metrics.
We have clarified this point in the “Strengths, limitations, practical applications, and future directions” section and acknowledge it as a limitation, in page 12, paragraph 2, 427-445 line. To facilitate interpretation, we report effect sizes with 95% confidence intervals for all internal load outcomes and suggest that these analyses be considered exploratory. Future studies with larger samples or endpoint-specific power calculations would help provide more definitive insights into internal load responses under hypoxic exposure.
- The authors described the cimcumstances of the testing taking place under controlled conditions. However, they would appear to be quite different regarding environmental conditions during the actual XC competitions, e.g. a XC race in cold environment, with wind chill effects, et cetera. I would therefore like to ask the authors to a) more clearly state these conditions during the testing, such as temperature, humidity, wind, et cetera, and b) to more thoroughly discuss the impact of possible differences of those conditions to actual XC competitions.
Response 3: a) We have applied the requested changes regarding laboratory conditions starting from page 3, paragraph 5, line 154.
- b) We have included a more comprehensive discussion of how potential differences in these conditions could affect actual XC competitions (see page 10, paragraph 2, line 332).
- The tables are a biut cumbersome to read (format and lots of information in certain cells), perhaps there is room for improvement.
Response 4: The tables have been reformatted and presented with the intention of improving the visibility and readability of the content.
- Figure 1 shows the mentioned correlations, however, 50% of the data shown is superfluous, since the correlation between for example Slope OBLA4 and Heart Rate OBLA4 is the same as between Heart Rate OBLA4 and Slope BBLA4. Furthermore, it should be clear that each parameter shows a correlation coefficient of 1 with itself, so I would suggest to delete these data.
Response 5: Autocorrelations have been removed, and the presentation structure of the graph has been improved. Changes are available in Graph 1.
- In line 298 that authors mention that the test subjects trained and competed at altitudes between PreCs and PosCs. Are there more deatiled descriptions regarding the altitude and the durantion during which the subject was training and competing under these conditions?
Response 6: in line 312 (number updated after incorporating previous corrections), descriptive data on training and competition, in terms of hours and sessions, have been added. In addition, we have addressed this as a study limitation in the 'Strengths, Limitations, Practical Applications, and Future Directions' section, paragraph 2, line 427
- In general, with regards to fatigue, I would suggest to the authors to consider and discuss also mental/psychological causes for fatigue (e.g. central governor theory), especially when discussing the changes from the pre to post competion phases.
Response 7: On page 9, fifth paragraph, starting at line 308, the potential explanation for fatigue related to the non-significant improvements after the competitive period (within the same testing condition) has been expanded to include the psychophysiological component, which could have affected the subjects’ exercise capacity and performance in the study protocol. Additionally, the central governor theory of fatigue, along with prior exposure to environmental recognition, has been included. Final additional comment: orthography, missing spaces, and superfluous spaces have been corrected based on the suggestions provided (lines 24, 74, 244, 290, 377).
Reviewer 2 Report
Comments and Suggestions for Authors
Delete “” in the title.
No headings are needed in the abstract, see author guidelines.
L6. Short title is not needed. Please delete, see author guidelines.
Please provide a reference(s) to justify the first statement on the role ofenergy production and movement efficiency.
Please consult author guidelines for referencing format.
L53. Are these references really needed. In addition, Shang et al is on skating performance. Sandbakk and Holmberg and Shang et al are experimental studie sbut you describe here what XC is all about. Any useful websites from XC organisations that would be better for referencing. Same argument for use of Sandbakk and Holmberg in L56.
L56. What is “m.a.s.l)”.
The introduction is focused on XC performance at sea level, it seems. Please provide justification why it would be of interest to examine under hypoxic conditions. Are there events at altitude that is examined in the present study? Is performance affected at 1800 m as that is mentioned in L56.
Ls 92-96. “performance variables” is not specific. Please be more precise.
L101. Please change “62.16 ± 11.1” to “62.2 ± 11.1” or even “62 ± 11”.
L100. Please provide age and height for the nine that completed the study.
L128. How long were the participants breathing normoxic or hypoxic air before testing. Why was there no measurement of arterial oxygen saturation. Is that a limitation?
Ls 138-139. Font is different.
L145. How well did the polynomials describe the data. Did you calculate correlation coefficients?
L158. “. Confidence intervals (CI) at 95% were provided where applicable”. Applicable for what. Please clarify.
Table 1. Provide heart rate values without decimal places, i.e. mean, SD, 95%CI etc. Also for Table 2.
Tables 1 and 2. Please ensure consistent number of decimal places for for example lactate values, as there are values with one decimal place and two decimal places.
L182. “percentaje”.
With the statistical analysis you can analyse differences but not absolute value pre and post for both conditions. Please get statistical advice.
The correlations in Figure 1 should be named a Table. Please do not present the correlations of 1 between identical parameters.
L203. “similar correlations were present”. Please revise as it seems to make a comparison with hypoxia.
L222. Check the manuscript for “hear” that needs to be “heart”.
L268. Do you have information on the number of races in the season? In addition, how long after the last race was the testing in the lab.
Ls 351-353. “In addition, significant correlations found after the training and competitive periods could help explain the specific enhancement of performance related to environmental conditions in training and competitions in XC skiing.” No information is provided on enhanced performance. Please clarify/revise.
Please consult author guidelines to provide consistent referencing format. For example, journal names are provide with full names and abbreviations.
Comments on the Quality of English LanguageThe manuscript, especially the discussion, is very challenging to read and comprehend. I suggest to revise for clarity.
Author Response
Dear Reviewer 2
Thank you very much for your insightful comments and feedback on our study. We truly appreciate your recognition of the value of our work, and we are grateful for the opportunity to improve the manuscript.
In response to your points, we have made the necessary revisions and incorporated the changes accordingly. Below, we present the modifications we have introduced in the manuscript for your review.
- Delete “” in the title.
No headings are needed in the abstract, see author guidelines.
L6. Short title is not needed. Please delete, see author guidelines.
Response 1: The quotation marks in the title have been removed, as requested. The heading in the abstract has also been deleted, and the short title has been removed in accordance with the author guidelines.
- Please provide a reference(s) to justify the first statement on the role ofenergy production and movement efficiency.
Response 2: reference has been added to support the initial statement rega
rding the role of energy production and movement efficiency.
- Please consult author guidelines for referencing format. L53. Are these references really needed. In addition, Shang et al is on skating performance. Sandbakk and Holmberg and Shang et al are experimental studie sbut you describe here what XC is all about. Any useful websites from XC organisations that would be better for referencing. Same argument for use of Sandbakk and Holmberg in L56.
Response 3: Thank you for the valuable suggestion. The revised version is much clearer than the original and enhances the overall understanding of the point. This change will be incorporated into the updated document. References concerning race distances have been revised following the suggestion to include official competition formats based on the International Competition Rules. The updated citation now reflects the official FIS regulations regarding cross-country skiing event types and distances.
Additionally, the previous experimental studies by Sandbakk (2017) and Shang (on skating performance determinants) have been replaced with appropriate references that focus on the current understanding of performance-determining factors specific to the competition format. This has been added from line 53 of the revised manuscript.
- What is “m.a.s.l)”.
Response 4: At line 53 term m.a.s.l is explained as meters above sea level.
- The introduction is focused on XC performance at sea level, it seems. Please provide justification why it would be of interest to examine under hypoxic conditions. Are there events at altitude that is examined in the present study? Is performance affected at 1800 m as that is mentioned in L56.
Response 5: On page 2, in paragraph 5, starting from line 80, the detrimental effect of altitude exposure on performance has been discussed. Additionally, in response to Reviewer 1's comments, the influence of altitude and its classification has been explored further. Due to the water vapour content at a given temperature, based on the Magnus-Tetens principle, the total air pressure can be calculated. From this, the actual PiOâ‚‚ at a given altitude can be estimated, considering the higher core body temperature during exercise. In this way, the higher apparent altitude used in the study is centred on the estimated PiOâ‚‚ under intense training and/or competition.
- Ls 92-96. “performance variables” is not specific. Please be more precise.
Response 6: thank you for the thoughtful feedback. Upon review, the original presentation of the idea was less clear, and your suggestion significantly improved its clarity. The input will be incorporated to enhance the presentation's effectiveness. The revised text now clearly refers to blood lactate concentration, heart rate, and external load expressed as slope, as shown in lines 92–96. In page 3, paragraph 3, line 110.
- Please change “62.16 ± 11.1” to “62.2 ± 11.1” or even “62 ± 11”.
Response 7: The value has been rounded accordingly. In line 109, the original value “62.16 ± 11.1” (previously in line 101) has been changed to “62.2 ± 11.1” to improve clarity and consistency in the presentation of results.
- Please provide age and height for the nine that completed the study.
Response 8: Thank you for the valuable feedback regarding the separate expression of weights and ages. The data on age and height for men and women, which completes the study, is presented on page 3, paragraph 3, lines 112-113.
- How long were the participants breathing normoxic or hypoxic air before testing. Why was there no measurement of arterial oxygen saturation. Is that a limitation?
Response 9: We have added that the participants were breathing hypoxic air for a 30-minute period prior to measurements to allow acclimatisation to the basal state under hypoxia exposure, rather than normoxia, and have included a supporting reference. Pag 3, paragraph 5, from line 135.
Battisti, A., Fisher, J. A., & Duffin, J. (2010). Measuring the hypoxic ventilatory response. Advances in experimental medicine and biology, 669, 221–224. https://doi.org/10.1007/978-1-4419-5692-7_44
Thank you for the valuable feedback regarding the use of arterial oxygen saturation monitoring. Typically performed on the participants' index finger with a pulse oximeter, this procedure was not feasible due to the motor tasks involved in the study. Since participants were handling implements with their hands, applying the device was not possible. Additionally, the decision was made to exclude this data to minimise any potential discomfort for the athletes. The study was designed to be as ecological as possible, aiming to be minimally invasive to keep the athletes as relaxed and natural as possible during the tests. Moreover, including saturation levels could have potentially provided cues to the athletes, which might have compromised the masking effect we aimed to achieve.
- Ls 138-139. Font is different.
Response 10: on page 4, paragraph 3, line 175 (previously lines 138–139), typographical errors have been corrected.
- How well did the polynomials describe the data. Did you calculate correlation coefficients?
Response 11: Regarding the polynomial model used in line 178 (previously 145): we recognise the importance of quantifying the goodness of fit when employing polynomial models. However, due to the inherent individual variability in lactate responses among participants, we applied a data-normality approach to ensure that the selected polynomial fitting method aligned with the expected data distribution. Specifically, we used 1.96 times the standard deviation (SD) to represent the 95% confidence interval of the measured data, thereby accounting for individual differences. We used boxplots for data visualisation to examine the distribution and variability of lactate and heart rate values across participants. This graphical approach further supports the suitability of the polynomial fitting method, rather than relying solely on correlation coefficients or R2 values, which may not fully capture the complexity of biological variability in this context.
Hopkins, W. G., Marshall, S. W., Batterham, A. M., & Hanin, J. (2009). Progressive statistics for studies in sports medicine and exercise science. Medicine and science in sports and exercise, 41(1), 3–13. https://doi.org/10.1249/MSS.0b013e31818cb278
- “. Confidence intervals (CI) at 95% were provided where applicable”. Applicable for what. Please clarify.
Response 12: On line 203 (previously 158), we clarified the statement regarding the use of 95% confidence intervals (CI) in the revised manuscript. Specifically, confidence intervals were provided for variables where estimating variability and precision was relevant, such as lactate concentration and heart rate measurements. This approach enables a more accurate interpretation of the physiological data and helps reflect the inherent variability among participants.
.
- Table 1. Provide heart rate values without decimal places, i.e. mean, SD, 95%CI etc. Also, for Table 2.
Response 13: We have removed the decimal places from the heart rate values in both Table 1 and Table 2. The means, standard deviations (SD), and 95% confidence intervals (CI) are now presented as whole numbers for clarity and consistency.
- Tables 1 and 2. Please ensure consistent number of decimal places for for example lactate values, as there are values with one decimal place and two decimal places.
Response 14: We have corrected the inconsistency in the number of decimal places in Tables 1 and 2 for heart rate. Lactate values and other relevant variables are now presented with a consistent number of decimal places throughout both tables.
- “percentaje”.
Response 15: The correction has been made as “percentage”; what was previously on line 182 is now on line 217 in the revised manuscript.
- With the statistical analysis you can analyse differences but not absolute value pre and post for both conditions. Please get statistical advice.
Response 16: we would like to clarify that the Wilcoxon signed-rank tests were applied to compare absolute values within each time point (pre or post) between conditions, as well as to assess differences between pre and post within each condition separately. However, we did not perform a combined analysis of interactions between condition (normoxia vs hypoxia) and time (pre vs post), which would require a different statistical approach.
This decision was made because, with the small sample size, applying statistical methods to examine interactions between condition and time could lead to Type I errors (false positives), which we aimed to avoid. Due to the risk of overinterpreting the data with such a limited sample, we chose a more conservative approach by analysing the pre- and post-conditions separately for each treatment. We sincerely appreciate your valuable feedback on this matter. It offers an interesting perspective that we will definitely consider in future studies, especially when larger sample sizes are available.
- The correlations in Figure 1 should be named a Table. Please do not present the correlations of 1 between identical parameters.
Response 17: We have modified the presentation accordingly in Figure 1: We have removed the autocorrelation values of 1 between identical parameters to avoid redundancy.
- “similar correlations were present”. Please revise as it seems to make a comparison with hypoxia.
Response 18: The sentence at line 203 (now line 238) has been revised for clarity. The phrase “similar correlations were present” has been modified to avoid implying a direct comparison with hypoxia, ensuring the statement accurately reflects the data presented.
- Check the manuscript for “hear” that needs to be “heart”.
Response 19: The typo “hear” has been corrected to “heart” throughout the manuscript. The correction is now reflected at line 258-259 (previously line 222).
- Do you have information on the number of races in the season? In addition, how long after the last race was the testing in the lab.
Response 20: We have now included information on the number of specific training hours and races during the season. These details have been added at line 307. The time elapsed between the last race and the laboratory testing was already specified in the Procedure section, page 3, paragraph 4, line 121.
- Ls 351-353. “In addition, significant correlations found after the training and competitive periods could help explain the specific enhancement of performance related to environmental conditions in training and competitions in XC skiing.” No information is provided on enhanced performance. Please clarify/revise.
Response 21: we have revised the statement for clarity on line 351-353. The sentence now reads: “In addition, significant correlations found after the training and competitive periods could help explain the specific changes in performance variables under environmental conditions of reduced PiO2 during a graded exercise test (GXT) after the training and competition period in XC skiing.” This revision clarifies that the correlations relate to changes in performance variables rather than generalised enhanced performance (which did not occur).
We have revised the referencing style throughout the manuscript to ensure consistency and compliance with the journal’s author guidelines. Specifically, journal names are now provided with full names or standard abbreviations as required by the journal’s format.
Additional language comments: we have thoroughly reviewed and revised the manuscript, particularly the Discussion section, to improve clarity and readability. We believe the revised version better conveys the key messages and is easier to follow.
Reviewer 3 Report
Comments and Suggestions for Authors
have provided all the comments to the authors in PDF format.

Author Response
To Reviewer 3
Thank you very much for your insightful comments and feedback on our study. We truly appreciate your recognition of the value of our work, and we are grateful for the opportunity to improve the manuscript.
In response to your points, we have made the necessary revisions and incorporated the changes accordingly. Below, we present the modifications, we have introduced in the manuscript for your review.
- Modify title, heading, short title.
Response 1: title, headings, and short title have been deleted.
- In the introduction, too much space was devoted to general information about cross-country skiing. Please emphasize why there is a need for this study. Toward the end of the introduction, highlight whether there are the same or similar studies that have addressed this topic. If such studies exist, explain them. If not, then make sure to point that out.
Response 2: In conjunction with feedback from other reviewers, we have provided a detailed justification for studying the impact of the hypoxic environment caused by altitude. Additionally, we included a classification of hypoxia in the Introduction to better contextualise the study. We also revised the Introduction to reduce the general information about cross-country skiing and to emphasise the specific need for this research. Towards the end of the Introduction, we clearly highlight whether similar studies exist. Where relevant, existing research is explained; where such research is lacking, this gap is explicitly stated to emphasise the originality and importance of our work. The new information can be found in lines 81, 84, and 87 to 98 of the revised document manuscript.
- Include reference to statement.
Response 3: references on line 47 related to “energy production and movement efficiency being critical determinants of performance” have been added.
- Response 4: We would like to note that, due to an error in the file format conversion process when submitting to the journal's editorial management system, the MDPI numerical citation system was not correctly maintained in the final manuscript. We apologise for any confusion this may have caused and appreciate your understanding.
- Lactate is an indicator of internal load, while external load refers to the prescribed work or exercise performe.- Please correct and provide a more adequate explanationd.
Response 5: The statement has been corrected and clarified on line 65 of page 2 (previously line 69). Specifically, we have specified the external load achieved at the mentioned lactate threshold.
- Replace “aerobic power” to “aerobic capacity”.
Response 6: aerobic power on line 71 (previously 77) has been corrected.
- Including hypothesis.
Response 7: It is acknowledged that the study initially lacked an explicit hypothesis. This has now been addressed, and the hypothesis has been added in the Introduction (page 2, line 4) to provide a clearer framework for the study. Thank you for bringing this to our attention. The inclusion of a hypothesis now ensures a more structured and focused approach to the research, and we appreciate your valuable feedback.
- The mean age and body mass should be presented separately for males, separately for females, and overall, for all participants
Response 8: The mean age and body mass have been reported separately for males, females, and the overall sample. This information has been added on page 3, paragraph 3, line 110 of the revised manuscript.
- Replace “threshold” for “level”.
Response 9: on page 5, paragraph 1, line 207 “level” has been changed. “The statistical significance level was set at p ≤ 0.05”.
- In the Statistical Analyses subsection, you did not mention regression analysis anywhere.
Response 10: Thank you for your feedback. It was our mistake during the formatting of the article that this information was accidentally removed. The regression analysis has now been included and can be found on pages 4–5, paragraph 5, line 190, in the Statistical Analyses subsection.
- At the beginning of the discussion, highlight the main findings of the study. After that, confirm the hypothesis that you defined as the last sentence in the introduction. Then, emphasize whether the results are in line with your expectations or not. If they are, explain why; if not, explain why not. Also, provide clearer explanations of the obtained results in the subsections: 4.1, 4.2, 4.3, and 4.4
Response 11: we have included the appropriate content regarding the study's objectives and hypotheses on Introduction section, page 3, paragraph 2, lines 100-113 according to these changes we have addressed the results based on hypothesis at the beginning of Discussion section, page 9, paragraph 4-6, lines 294-321. Between lines 348–368 and 391-399 we have addressed the comparison of the evidence found in the study. This includes contrasting the effects of altitude exposure combined with other stressors, as well as discussing the potential applications and implications of the results.
- I do not understand at all why, throughout the text, blood lactate concentration and HR are referred to as external load. These are internal load variables. Variables for external load include, for example: speed, slope/incline, distance, power/output, and exercise duration.
Response 12: Corrections have been made on page 9, paragraph 4, line 289. We have clarified in the manuscript that no significant differences were detected in internal load variables (blood lactate concentration and heart rate), as well as in external load measures. We appreciate the reviewer’s careful observation and have adjusted the terminology accordingly.
- Response 13: the word “studies” has been corrected on line 402 (previously line 350).
- Response 14: section “Strengths, limitations, practical applications, and future directions” has been added.
Many kind regards.
Round 2
Reviewer 2 Report
Comments and Suggestions for Authors
Thanks for responding to my comments and suggestions. There is still one regarding data presentation.
If you express heart rate values without decimal places for the mean and SDs, as you should do, then that applies as well to 95% CI and diff values of the heart rate observations. Similar reasoning for other parameters in Tables 1, 2 and 3. Please revise.
Author Response
Comment 1: If you express heart rate values without decimal places for the mean and SDs, as you should do, then that applies as well to 95% CI and diff values of the heart rate observations. Similar reasoning for other parameters in Tables 1, 2 and 3. Please revise
Response 1: Thank you very much for the valuable observations and for the advice regarding the unification of statistical results. We have implemented the modifications in the heart rate values for the confidence intervals and the difference values. In addition, due to rounding to whole numbers, adjustments have also been made in the classification based on effect size. Changes are presented in Results section: Table 1, Table 2 and Table 3.
Furthermore, the number of decimal places has been adjusted according to the magnitude of the differences and the confidence intervals for other variables. These revisions have been consistently applied across Tables 1, 2, and 3, as well as on page 7, lines 248 to 263. In these lines, two decimal places for correlations have been maintained to ensure the statistical results are as precise and comprehensible as possible.
We hope that these revisions address your suggestions satisfactorily, and we remain grateful for your insightful feedback.
Kind regards.
Reviewer 3 Report
Comments and Suggestions for Authors
Dear,
The authors have made revisions in accordance with my comments and suggestions, and I now consider the manuscript ready for publication. The only change needed is in the discussion section, where two words should be modified for a more precise interpretation of the term.
Kind regards.

Author Response
Comment 1: The authors have made revisions in accordance with my comments and suggestions, and I now consider the manuscript ready for publication. The only change needed is in the discussion section, where two words should be modified for a more precise interpretation of the term.
Response 1: Thank you very much for your valuable observations and for your advice regarding the suggested word changes.
The changes have been applied in the discussion section (page 9, paragraph 4, lines 294 and 302), where two words need to be modified for a more precise interpretation of the term.
We hope that these revisions address your suggestions satisfactorily, and we remain grateful for your insightful feedback.
Kind regards.